# Surface Albedo and Snowline Altitude Estimation Using Optical Satellite Imagery and In Situ Measurements in Muz Taw Glacier, Sawir Mountains

**Fengchen Yu** [1,2]**, Puyu Wang** [1,2,]*** and Hongliang Li** [1]

1   State Key Laboratory of Cryosphere Science, Northwest Institute of Eco-Environment and Resources, Chinese Academy of Sciences, Lanzhou 730000, China
2   College of Sciences, Shihezi University, Shihezi 832000, China
*   Correspondence: wangpuyu@lzb.ac.cn

**Abstract:** Glacier surface albedo strongly affects glacier mass balance by controlling the glacier surface energy budget. As an indicator of the equilibrium line altitude (ELA), the glacier snowline altitude (SLA) at the end of the melt season can reflect variations in the glacier mass balance. Therefore, it is extremely crucial to investigate the changes of glacier surface albedo and glacier SLA for calculating and evaluating glacier mass loss. In this study, from 2011 to 2021, the surface albedo of the Muz Taw Glacier was derived from Landsat images with a spatial resolution of 30 m and from the Moderate Resolution Imaging Spectroradiometer albedo products (MOD10A1) with a temporal resolution of 1 day, which was verified through the albedo measured by the Automatic Weather Station (AWS) installed in the glacier. Moreover, the glacier SLA was determined based on the variation in the surface albedo, with the altitude change along the glacier main flowline derived from the Landsat image at the end of the melt season. The correlation coefficient of >0.7, with a risk of error lower than 5%, between the surface albedo retrieved from remote sensing images and the in situ measurement data indicated that the method of deriving the glacier surface albedo by the remote sensing method was reliable. The annual average albedo showed a slight upward trend (0.24%) from 2011 to 2021. A unimodal seasonal variation in albedo was demonstrated, with the downward trend from January to August and the upward trend from August to December. The spatial distribution of the albedo was not entirely dependent on altitude due to the dramatic effects of the topography and glacier surface conditions. The average SLA was 3446 m a.s.l., with a variation of 160 m from 2011 to 2021. The correlation analysis between the glacier SLA and annual mean temperature/annual precipitation demonstrated that the variations of the average SLA on the Muz Taw Glacier was primarily affected by the air temperature. This study improved our understanding of the ablation process and mechanism of the Muz Taw Glacier.

**Keywords:** glacier surface albedo; snowline altitude; spatio-temporal variations; Muz Taw Glacier; Sawir Mountains

## 1. Introduction

Under the background of climatic warming, most global glaciers have shown a state of mass loss since the 1950s [1–3]. The accelerated mass loss of glaciers significantly affects river runoff and the regional water cycle, as well as the ecology and social economy, and increases the possibility of regional water resource disasters [4–6]. As a critical component of the cryosphere, mountain glaciers are sensitive to subtle changes in the climate and play a pivotal role in regional ecological development and hydrological cycles [7–9]. Especially in arid and semi-arid regions, glaciers are extremely important solid water resources. For the Sawir Mountains, glaciers are not only important freshwater resources for the production and life of residents in Jimunai County of the Altai region in Xinjiang Uygur Autonomous Region and its surrounding areas, but they also provide the water supply for regional

rivers, such as Ulequin Urastu River and Ulast River. According to previous studies [10,11], glaciers in the Sawir Mountains have experienced accelerated retreat since 1959, with the area shrinkage rate larger than 40%.

As an important factor controlling the glacier surface energy budget, the glacier surface albedo is the ratio of total solar radiation reflected by the glacier surface to the total solar radiation received. Meanwhile, the glacier surface albedo plays a key role in modulating glacier melt, the variability of which depends on numerous complex factors, such as air temperature, precipitation, topography, cloud cover, and light absorbing impurities [12–14]. The glacier surface energy balance is extremely sensitive to variations in albedo, which further affects the glacier mass balance by controlling net shortwave radiation on the glacier surface [15–17]. In addition, the ELA refers to the altitude where the annual glacier ablation equals the accumulation in a hydrological year [18]. The SLA is considered to be the boundary between snow and bare ice. The SLA approximates the ELA at the end of the glacier melt season, which can be used to reflect the glacier mass loss or accumulation [19–21].

Remote sensing has become an effective method to study glacier variation because of its high temporal and spatial resolutions and wide region coverage. Generally, glacier surface albedo and SLA are investigated based on Landsat images with higher spatial resolution and Moderate Resolution Imaging Spectroradiometer (MODIS) products with higher temporal resolution [22–24]. Although Landsat images are more susceptible to cloud cover and shadow, the temporal resolution of which is lower than that of MODIS images, the combination of Landsat images and MODIS can be used to obtain the surface albedo with a higher spatial resolution and a longer time series. A series of Landsat images in the melt season are ideal for determining the glacier SLA. The glacier SLA can be extracted by calculating the band ratio of the satellite image [25,26] or by classifying the spectral reflectance map according to spectral mixture analysis, supervised classification, unsupervised classification, or decision-tree methods [27,28]. The glacier surface albedo variations with altitude can effectively reflect the change of physical composition of the glacier surface. Thus, it is feasible to determine the glacier SLA by the difference in albedo between ice and snow derived from Landsat images at the end of the melt season [29–31].

Taking this into account, in this study, Landsat images, MOD1OA1 albedo products from 2011 to 2021, and in situ measurements by the AWS on the Muz Taw Glacier in the Sawir Mountains were firstly used to investigate the variations of glacier surface albedo and the contribution of surface albedo to glacier melt. Then, the glacier SLA was extracted based on the surface albedo along the glacier main flowline derived from Landsat images at the end of the melt season. The glacier SLA variations and the influence of climate variables on SLA are also assessed. Our purpose is to (i) evaluate the multi-scale surface albedo variation and understand the feedback mechanism of albedo on glacier mass balance and (ii) determine glacier SLA from Landsat images to reveal the trend of glacier SLA and its sensitivity to climate variables.

## 2. Study Area

The Altai Mountains is a transnational mountain range in east-central Asia bordering China, Russia, Kazakhstan, and Mongolia. Friendship Peak (4374 m a.s.l.) is the highest peak in the Altai Mountains, and it is located in the upper reaches of the Buerjin River at the junction of China and Mongolia. There were a total of 416 glaciers with an area of 293.2 km$^2$, according to the first Glacier Inventory of China (GIC-1) [32]. Located in the transitional section between the Tianshan Mountains and the central Altai Mountains, glaciers exist in the Sawir Mountains spanning China and Kazakhstan. A total of 21 glaciers with an area of 16.84 km$^2$ were distributed in the Sawir Mountains, according to GIC-1 [32], including 13 glaciers on the northern side and 8 glaciers on the southern side of the Sawir Mountains, respectively.

The Muz Taw Glacier (47°04′N, 85°34′E) is a typical northeast-orientated valley glacier, and it is located on the northern side of the Sawir Mountains, with an area of 4.27 km$^2$ and a length of 3.7 km, according to GIC-1 (Figure 1) [33]. The melt water on Muz Taw

Glacier flows into the Ulequin Urastu River, which is a transboundary river between China and Kazakhstan. The Muz Taw Glacier has shrunk extremely due to dramatic melt in the past 30 years [33,34]. The glacier area decreased from 3.97 km$^2$ to 3.15 km$^2$ for the period from 1977 to 2013 [34]. The area of Muz Taw Glacier decreased to 3.13 km$^2$, with a length of 3.2 km in 2016 [35]. The Muz Taw Glacier is affected by the prevailing westerlies and the Asian anticyclone and polar air mass in winter [36]. The annual average temperature and annual precipitation in this region increased at a rate of 0.4 °C·10a$^{-1}$ and 12 mm·10a$^{-1}$ during 1961–2016, respectively [35].

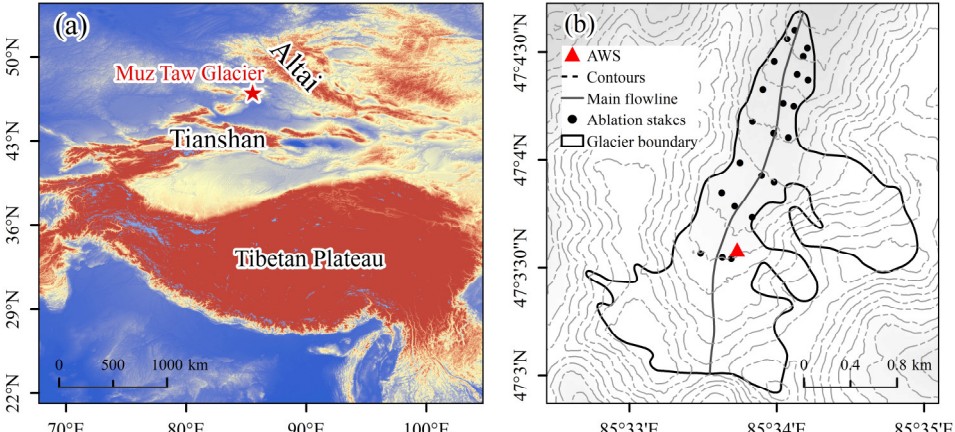

**Figure 1.** (**a**) Location map of the study area with the red five-pointed star representing the Muz Taw Glacier in the Sawir Mountains, Central Asia. (**b**) Topography of the Muz Taw Glacier, with the distribution of ablation stakes and the AWS.

## 3. Data and Methods

### 3.1. Data Sources

#### 3.1.1. Remote Sensing Data

In this study, Landsat-7 Enhanced Thematic Mapper Plus (ETM+) and Landsat-8 Operational Land Imager (OLI) images with a spatial resolution of 30 m were used to extract SLA. Landsat images were downloaded from the USGS (https://earthexplorer.usgs.gov/, accessed on 4 August 2022) and Geospatial Data Cloud (http://www.gscloud.cn/, accessed on 4 August 2022) (Table 1). Because the glacier melt generally occurs from June to August, and occasionally in early September, the data time series, fresh snowfall, and cloud cover should be considered when selecting Landsat satellite images. Therefore, the images were selected during the melt season (usually from June to early September) to ensure high quality and avoid the impact of the cloud cover in the glacier area. Moreover, images with fresh snowfall events at the moment of satellite transit were excluded. Finally, 13 Landsat images from 2011–2021 were screened, and images from 7 July to 16 September were used to determine the glacier SLA. However, the Landsat ETM+ instrument equipped with a scan line corrector (SLC) failed after May 2003, resulting in data gaps. The glacier in the study area is completely blocked by cloud cover or temporary snowfalls at the end of the melt season, resulting in missing data (2017) during the study period.

The Moderate Resolution Imaging Spectroradiometer products (MOD10A1) with the high temporal resolution of 1 day were used to analyze the multi-temporal variations of glacier surface albedo in this study. MOD10A1 was obtained from the National Snow and Ice Data Center (NSIDC, http://nsidc.org/, accessed on 4 August 2022).

**Table 1.** Landsat images used to determine the SLA on the Muz Taw Glacier in the Sawir Mountains during the period 2011–2021.

| Path/Row | Date | Sensor | Cloud Cover |
|---|---|---|---|
| 145/27 | 12 July 2011 | Landsat ETM+ | 1% |
| | 16 September 2012 | Landsat ETM+ | 5% |
| | 3 September 2013 | Landsat ETM+ | 0 |
| | 29 August 2014 | Landsat OLI | 0.13% |
| | 15 July 2015 | Landsat OLI | 0.02% |
| | 26 August 2016 | Landsat ETM+ | 0 |
| | 5 June 2018 | Landsat OLI | 1.2% |
| | 7 July 2018 | Landsat OLI | 0.93% |
| | 27 August 2019 | Landsat OLI | 0.02% |
| | 14 October 2019 | Landsat OLI | 0.25% |
| | 9 May 2020 | Landsat OLI | 0.54% |
| | 14 September 2020 | Landsat OLI | 1.9% |
| | 24 August 2021 | Landsat ETM+ | 0 |

The Digital Elevation Model (DEM) was used to assist the topographic correction for the Landsat images and determine the specific altitude of the snowline. ASTER GDEM v3 was chosen and downloaded from the Geospatial Data Cloud (http://www.gscloud.cn/, accessed on 4 August 2022) with a spatial resolution of 30 m. The horizontal and vertical accuracies of the DEM were 30 m and 20 m, respectively.

### 3.1.2. In Situ Measurements

An automatic weather station (AWS) has been installed at the altitude of 3430 m a.s.l. on a relatively flat surface near the equilibrium line altitude (ELA) of the Muz Taw Glacier since 2018. The CNR4 pyranometer is mounted on the AWS at a height of 1.5 m. The in situ-measured albedo was obtained by calculating the ratio of reflected and incoming shortwave radiation observed from the CNR4 pyranometer, with an accuracy of albedo less than 0.01 in the wavelength range of 0.3~2.8 um. The measured albedo from 2018 to 2021 were selected to verify the accuracy of surface albedo derived from remote sensing images.

The mass balance of the Muz Taw Glacier had been measured by ablation stakes and snow pits since 2015. A total of 22 ablation stakes were drilled into the glacier using a steam drill and distributed at different altitudes. The measurement of mass balance includes the thickness and density of each snow layer at snow pits, the vertical height from the top of the ablation stakes to the glacier surface, and the thickness of superimposed ice. The mass balance of each measured point can be calculated as the arithmetic sum of the snow, glacier ice, and superimposed ice mass balance. Point values are extrapolated to glacier-wide mass balance using the contour-line or profile method. Measured glacier-wide mass balances from 2017 to 2021 were used to analyze the potential impact of surface albedo on glacier melt in this study.

### 3.1.3. Meteorological Data from ERA-5

ERA 5 atmospheric reanalysis data were downloaded from the European data center (http://apps.ecmwf.int/datasets/, accessed on 4 August 2022). In this study, we selected monthly average air temperature and precipitation data with a spatial resolution of 0.1° during 2011–2021 for analyzing the trend of temperature and precipitation on the Muz Taw Glacier and for discussing the effects of temperature and precipitation on the glacier SLA variations.

### 3.2. Methods and Data Processing

#### 3.2.1. The Glacier Surface Albedo Derived from Landsat and MOD10A1

The narrowband albedo received by Landsat images was calibrated to broadband albedo in order to better divide the boundary of ice and snow based on the albedo of the glacier surface. The procedures of albedo derived from Landsat images include ge-

olocation, radiometric calibration, atmospheric correction, topographic correction, and narrow-to-broadband. It should be noted that the Landsat ETM+ instrument with a scan line corrector malfunctioned in 2003, resulting in data gaps. These data gaps were repaired by neighboring pixels of the image gaps.

Different methods were used for radiometric calibration of Landsat ETM+ and Landsat OLI in order to perform atmospheric corrections [37].

For Landsat ETM+, the calculation equation is as follows:

$$L = \frac{(L_{max} - L_{min}) \times (DN - DN_{min})}{(DN_{max} - DN_{min})} + L_{min} \tag{1}$$

where $L_{min}$ and $L_{max}$ are the minimum and maximum spectral radiance band limit, $DN$ is the satellite image digital number, and $DN_{max}$ and $DN_{min}$ are the maximum and minimum satellite image digital number, respectively.

For Landsat OLI, the calculation equation is as follows:

$$L_\lambda = G \times DN + B \tag{2}$$

where $L_\lambda$ is the radiance at the sensor's aperture of the Landsat OLI band $\lambda$[W(m$^{-2}$·sr $^{-1}$·μm$^{-1}$)], $G$ is the gain of sensor [W$^{-1}$(m$^2$·sr·μm)], $DN$ is the satellite image digital number, and $B$ is the bias of the sensor.

The FLAASH (Fast Line-of-sight Atmospheric Analysis of Spectral Hypercubes) method was applied to eliminate the influence of atmospheric scattering, absorption, and reflection for the surface reflectivity [38]. Based on the solar radiation spectral range (excluding thermal radiation) and Planar Lambertia (or Approximate Planar Lambertia), the spectral radiance of pixels ($L$) received at the sensor was calculated as follows:

$$L = \left(\frac{AP}{1 - P_eS}\right) + \left(\frac{BP}{1 - P_eS}\right) + L_\propto \tag{3}$$

where $A$ and $B$ are coefficients depending on atmospheric and geometric conditions, and $P$ and $P_e$ are the surface reflectivity of pixels and the average reflectivity of pixels and the surrounding area, respectively. $S$ is the spherical albedo of the atmosphere, and $L_\propto$ is the radiance backscattered by the atmosphere, respectively.

The principle of the C-factor correction model is that there is a linear relationship between the pixel DN value of any band image and the cosine of the corresponding solar incidence angle [39]. The parameter C is the ratio of intercept $a$ to slope $b$ of the linear equation fitted by the pixel DN value and the cosine of the solar incidence angle. In order to avoid the complex process of using a large number of discrete samples to calculate the fitting coefficients $a$ and $b$, we used the improved C-factor correction model combined with the DEM data of the study area [40]. The expression is as follows:

$$L_h = (L_t - L_{min}) \times \frac{(cos\delta - cos\theta_{min})}{(cos\theta - cos\theta_{min})} + L_{min} \tag{4}$$

where $L_h$ is the radiation value of a point on the horizontal ground, $L_t$ is the radiation value of a point on the inclined ground, $L_{min}$ is the minimum $DN$ value in the shadow area on satellite images, $\delta$ is the solar zenith angle, and $\theta$ is the solar incident angle.

The conversion of linear equation from Landsat narrowband albedo to broadband albedo is used:

$$R = a + b_1CH_1 + b_2CH_2 + \dots b_nCH_n + \varepsilon \tag{5}$$

where $CH_1, CH_2, \dots, CH_n$ represent the specific band albedo of the satellite; and $a$ and $b_1, b_2, \dots, b_n$ represent the polynomial regression coefficients, which are determined by the surface and atmospheric conditions. However, due to the large temporal and spatial differences, this method is not universally applicable. Finally, we establish an equation for

narrow to broadband albedo from green, near-infrared, and thermal infrared bands based on Duguay et al. [41], and the specific conversion formula is given as:

$$\alpha = 0.526\alpha_g + 0.314\alpha_n + 0.112\alpha_t \tag{6}$$

where $\alpha_g$, $\alpha_n$, and $\alpha_t$ represent the albedo of the green band, NIR band, and thermal infrared band, respectively.

The surface albedo derived from the MOD10A1 product corresponds to the broadband albedo for the actual direct and diffuse illumination, and it has been subjected to atmospheric correction and anisotropy correction by the Discrete Ordinates Radiative Transfer (DISORT) model [42,43]. The pixel values ranging from 0 to 100 within the Muz Taw Glacier boundary were counted in this study.

### 3.2.2. Snowline Altitude Extraction

The glacier SLA was determined based on the surface albedo change along the glacier main flowline and the registered DEM [30,31]. The glacier main flowline was determined based on connecting the points with the maximum curvature on the contour line. The threshold of ice and snow based on the changes of surface albedo with altitude rising was defined to determine the contour of bare-ice or snow-covered surfaces. A 100 m wide glacier main flowline was extracted from the Landsat images at the end of the melt season. Contour lines were generated from the calibrated DEM at the altitude intervals of 50 m and 5 m. The extracted contours were superimposed with the derived albedo map of the main flowline to obtain the albedo of the glacier main flowline at 50 m intervals. The point with the largest standard deviation of albedo and where albedo begins to decrease continuously should be given special attention as both represent dramatic variations in the properties of the glacier surface in the corresponding altitude zone. Therefore, the two points were considered to be the albedo thresholds for distinguishing ice and snow, and the corresponding altitude zone was considered to be the range at which the snowline was located. However, SLA can probably run through several contour lines at the same time, and it was not evenly distributed according to the contour lines. Therefore, the 5 m interval contour layer was overlaid with the classified image obtained according to the albedo threshold. The average value of the contours closest to the boundary was taken as the final snowline altitude.

## 4. Results and Analyses

### 4.1. Long-Term Variation of Glacier Surface Albedo

As shown in Figure 2a, there were mostly positive anomalies from 2011 to 2021, and the magnitude of negative anomalies was weakened, indicating an increasing trend for annual average albedo, with an annual average albedo increase of approximately 0.17%. The average albedo also showed an increasing trend from May to August, with an annual average albedo increase of approximately 0.24%. The glacier surface albedo in 2012 was almost equal to the average albedo.

For the monthly scale, the glacier surface albedo indicated significant variation during 2011–2021 (Figure 2b). On the whole, the average albedo showed a decreasing trend from January to August and an increasing trend from August to December. The average glacier surface albedo was 0.47 during 2011–2021, which was used as a reference value for seasonal variations in albedo. A peak value of positive anomalies occurred in January, and then the magnitude of positive anomalies exhibited a dramatic decline from January to April. The magnitude of negative anomalies declined sharply between May and September, until it reached a peak value in August. The albedo anomalies returned to positive anomalies again from October to December because of the fresh snowfall and aged snow accumulation, with an accelerated upward trend. Generally, the minimum albedo appeared between late June and early September. The maximum albedo usually occurred from December to February as the fresh snow fell. The glacier surface albedo exhibited an evident seasonal evolution, which implied that the glacier surface absorbed more shortwave radiation or reflected less

shortwave radiation from June to early September, when there was lower albedo, and this was related to the variations of air temperature and precipitation during these months.

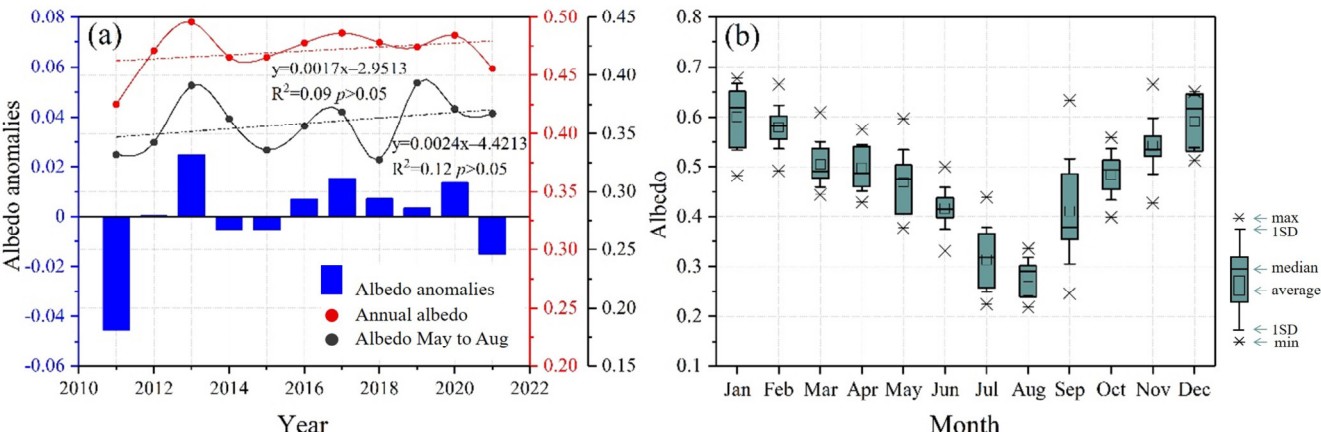

**Figure 2.** The annual (**a**) and the seasonal variation (**b**) in surface albedo on the Muz Taw Glacier during 2011-2021. The blue bars represent the annual average glacier surface albedo anomalies. The red and black dots indicate the annual average glacier surface albedo and the surface albedo during ablation period (May to August), respectively, and the dotted line represents linear trend.

### 4.2. Multi-Scale Variability of Glacier Surface Albedo

The glacier surface albedo variations during the melt season were particularly important for glacier melt. However, limited by the cloud cover or temporary snowfalls, the suitable Landsat images during the melt season within a year were not obtained. Therefore, variations in glacier surface albedo during the melt season (selected here from May to October) in this study were investigated qualitatively by Landsat images of the adjacent period from 2018 to 2020.

Figure 3 shows glacier surface albedo variations during the melt season from May to October. The average glacier surface albedo was 0.66 in May. The surface albedo was higher than 0.6 for approximately 78% of the glacier areas, and the proportion of glacier area accounted for 47% when the albedo was higher than 0.7. The average surface albedo was 0.63 in June, with most values (49%) ranging between 0.6 and 0.7. The average surface albedo was 0.43 in July. The proportion of areas with albedo less than 0.4 was 39%, with 43% of values ranging from 0.4 to 0.6, and only 18% of the areas higher than 0.6. The average albedo in August was 0.47, with 50% of the glacier surface albedo between 0.5 and 0.7. The average albedo was 0.65 in September and 0.56 in October, and most values (70%) were greater than 0.6 and most areas (69%) were greater than 0.5, with the largest number of pixels between 0.6 and 0.7 (30%), respectively. Overall, the glacier surface albedo would continue to decline throughout the melt season and eventually reach its minimum. Although the albedo continued to decline between May and June, the value remained above 0.6, and then the glacier surface albedo showed an accelerated decline in July, and stabilized at the minimum ordinarily between July and August, when the value remained below 0.5. The surface albedo increased again with the snowfall and fresh snow accumulation from late August or the beginning of September. These phenomena were also found on Urumqi Glacier No.1 in TianShan [23].

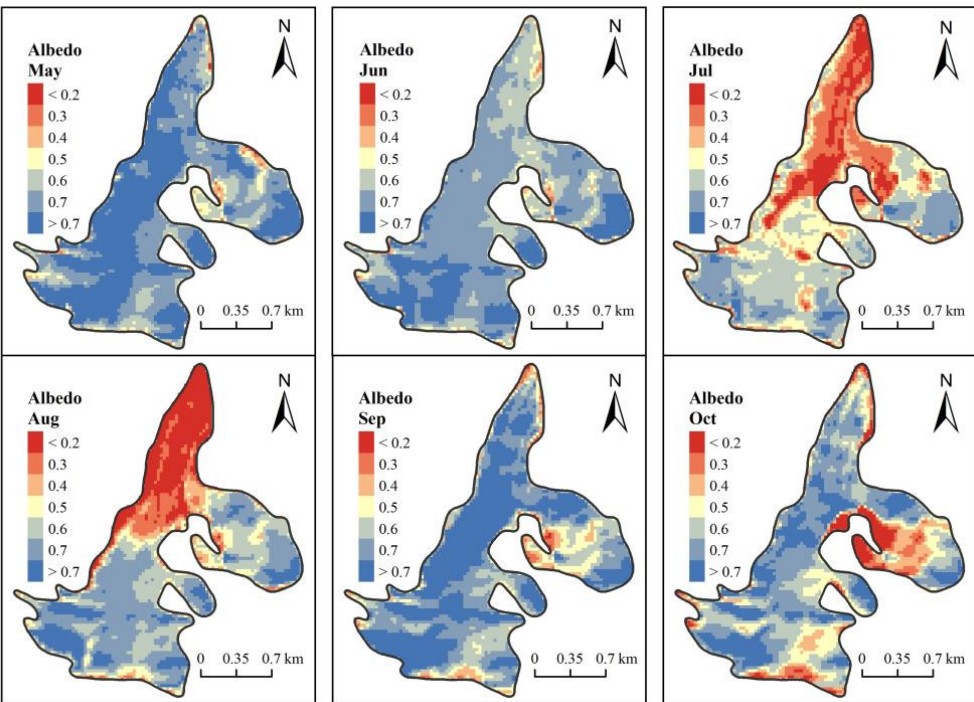

**Figure 3.** Spatial distribution of glacier surface albedo derived from Landsat images near the melt season in the different periods.

The spatial variation in the surface albedo on a single glacier was primarily affected by topographic factors. The albedo along the glacier main flowline can be used as an indicator to describe the spatial distribution of the glacier surface albedo. Thus, we investigated the albedo variations along the glacier main flowline at 50 m altitude intervals from May to October (Figure 4). The results displayed that the glacier surface albedo generally increased with rising altitude, and an upward, obvious, steep gradient occurred especially at the altitude range of 3350–3550 m a.s.l. along the glacier main flowline. However, the glacier surface albedo above 3600 m a.s.l. decreased obviously with rising altitude. The increase in the surface albedo was due to the weakening of the glacier melt, the reduction of the bare ice area, and the expansion of the fresh snow or firn area with rising altitude. The decrease in the surface albedo was probably due to the increase of the bare rock at the top of the glacier. Furthermore, the largely undulating terrain above 3600 m a.s.l. made it difficult to retain snow. The change of the solar incident angle caused by the slope and aspect was probably another reason for the obvious decrease of albedo in this altitude zone. It was obvious that the variation in the surface albedo with the altitude was more significant from July and August, further confirming that Landsat images at the end of the melt season should be given priority in extracting the glacier SLA, which will be discussed in detail in Section 4.3.

### 4.3. Variability of Glacier Snowline Altitude

The glacier SLA at the end of the melt season should generally be the highest SLA for a given year, meaning that it can be used to characterize the inter-annual variation trend of glacier SLA. As described above, the surface albedo generally increased along with the rising altitude. There are two trends in determining the albedo threshold. The first trend in surface albedo variation along with rising altitude is shown in Figure 5a, with two obvious inflection points. The first inflection point is located at an altitude of 3350–3400 m a.s.l. Because of the largest standard deviation of albedo at this altitude zone, the changes in the properties of cover objects on the glacier surface are most drastic in this altitude zone, which is defined as the boundary between the bare-ice area and the ice/snow transition area. The second inflection point is located at an altitude of 3500–3550 m a.s.l., as the surface

albedo began to decline, which is defined as the boundary between the ice/snow transition zone and the snow cover zone. The boundary between the ice/snow transition zone and the snow cover zone is also called the snowline, according to [30].

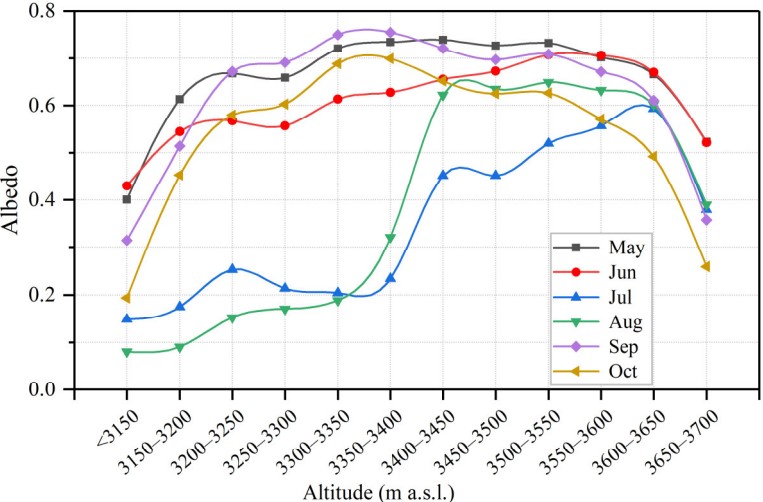

**Figure 4.** Variability in surface albedo along the glacier main flowline.

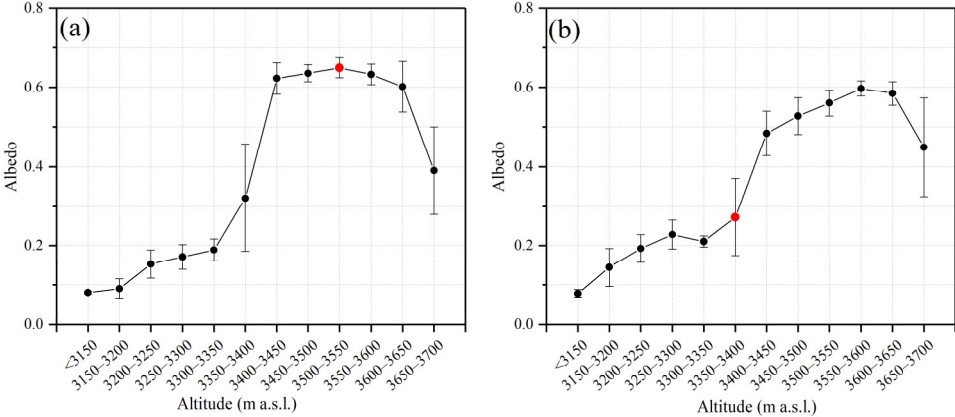

**Figure 5.** The average albedo variation of the 50 m altitude intervals along the glacier main flowline in (**a**) 2019, (**b**) 2015, respectively.

The second trend in surface albedo variation along with altitude rising is shown in Figure 5b. There is an obvious inflection point of the albedo, located at an altitude zone of 3350–3400 m a.s.l., because of the largest standard deviation. It is regarded as the boundary between the bare ice and the snow cover area. The albedo thresholds for other periods are determined based on the same principle. As shown in Figure 6, the SLA of the Muz Taw Glacier from 2011 to 2021 was finally obtained. The SLA of the glacier at the end of melt seasons from Landsat images was investigated, ranging from 3375 m a.s.l. in 2015 to 3535 m a.s.l. in 2021, with an average SLA of 3446 m a.s.l. and a variation of 160 m. A trend of continued melting was manifested on the Muz Taw Glacier during 2011–2021, with an increased rate of approximately 6.9 m·a$^{-1}$. However, the SLA in 2017 was not obtained because the Landsat image was completely blocked by temporary snowfalls or cloud cover in later summer.

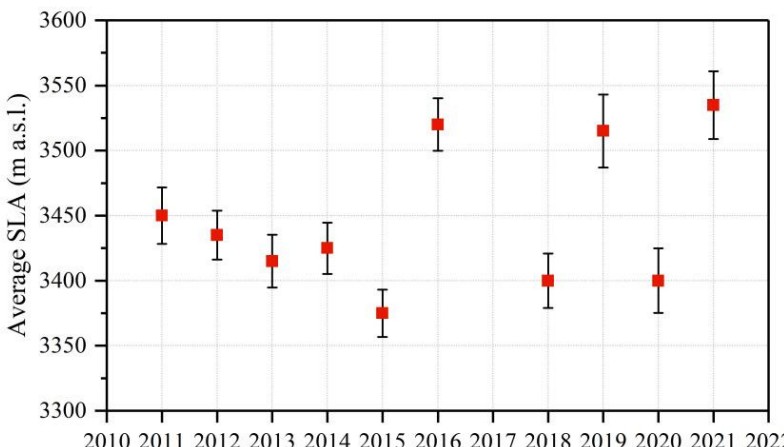

**Figure 6.** The average snowline altitude of the Muz Taw Glacier in the Sawir Mountains derived from Landsat over the period during 2011–2021.

## 5. Discussion

### 5.1. Uncertainty Estimation of Glacier Surface Albedo

The accuracy of the albedo derived from the satellite is affected by the temporal and spatial resolution of the satellite imagery, the length of the data time series, cloud cover, fresh snowfall, processing methods, etc. When selecting satellite images for this study, the impact of cloud and snowfall events was avoided as much as possible. In order to evaluate the impact of different data sources on the accuracy of the albedo, the albedo values retrieved from Landsat at the pixel where the AWS was installed on the glacier were compared with the albedo values measured by the AWS during the same period. The albedo measured by the AWS on 27 August 2019, 14 October 2019, 9 May 2020, and 17 September 2020, were used. The correlation was 0.95 at the 95% confidence level between the Landsat-derived and the measured albedo. The Absolute Error (AE) ranged from 0.01 to 0.07, with a Root Mean Square Error (RMSE) of 0.038 and a Mean Absolute Error (MAE) of 0.03 (Figure 7a).

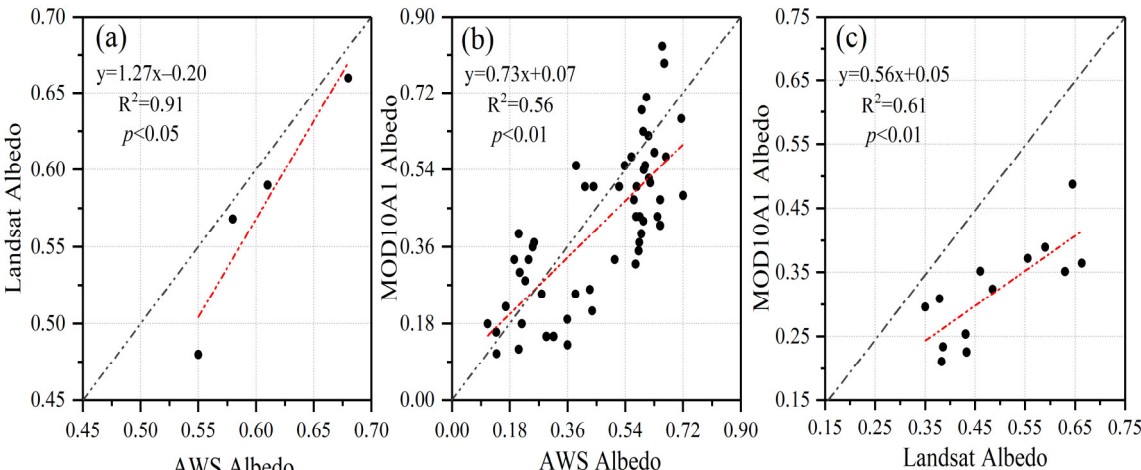

**Figure 7.** The scatter plots of the albedo values measured by the AWS against the values retrieved from Landsat images (**a**) and MOD10A1 (**b**). The relationship between the albedo derived from Landsat image and MOD10A1 albedo (**c**).

The MOD10A1 albedo values were also compared with the albedo measured in situ by the AWS from May to August 2020, which indicated that the correlation between the MOD10A1 albedo and the measured albedo was 0.75 at the 99% centroid level and that the difference ranged from −0.18 to 0.25, with an RMSE of 0.13 and an MAE of 0.05, respectively

(Figure 7b). The deviation between the MOD10A1 albedo and the measured albedo was larger than that of the albedo derived from Landsat images. The greater deviation in surface albedo derived from the MOD10A1 mainly stemmed from two aspects: (1) spectral properties, meaning the snow albedo was higher in the visible spectrum and lower in the shortwave and infrared bands; (2) mixed pixels, meaning the average from the 500 m pixel resolution of the MOD10A1 corresponds to the point-scale AWS data, and the biases induced by mixed pixels at the edge and outside were not considered.

In the current study, although there were fewer verification points for albedo derived from Landsat images, more AWS data were used to validate the accuracy of the MOD10A1. A significant agreement was observed between the albedo from MOD10A1 products and the in situ-measured albedo. Thus, the method recommended by Liang et al. [44] was adopted to further validate the accuracy of the surface albedo derived from Landsat images. The albedo derived from Landsat images were up-scaled and aggregated into a spatial resolution range of 500 m. The MOD10A1 albedo products were up-scaled and aggregated into a temporal resolution range of 15 d. As proposed by Gunnarsson et al. [45], "the temporal aggregation range was set as the number of days backwards and forwards at each center date (t = 0) to merge to a single stack for further processing". We selected a temporal aggregation range of 7 d backward and forward (t = ±7 d); in total, 15 d can contribute data to each center date albedo. In this way, the spatio-temporal resolution of the Landsat and MOD10A1 images were consistent in order to further validate the accuracy of the surface albedo. The correlation coefficient was 0.78 at the 99% confidence level between the two remote sensing products, i.e., Landsat and MOD10A1 (Figure 7c).

Although different data sources and processing methods can introduce some uncertainty in surface albedo estimation, as mentioned above, both remote sensing products can be used in combination to study the spatial and temporal variations of the surface albedo and the SLA. This study took full advantage of the high spatial resolution of Landsat images, the high temporal resolution of the MOD10A1, and the reliability of in situ measurement, which has wide application prospects for further research.

### 5.2. Uncertainty Estimation of Glacier Snowline Altitude

The uncertainty in the glacier SLA was estimated mainly from the precision of (1) DEM, (2) Landsat images, and (3) data processing steps. The vertical error due to DEM was about 20 m in this study area. Approximately 75% of the Landsat scenes were analyzed for glacier SLA extraction, with 54% acquired in July to August and 23% acquired in early September. Landsat images acquired at different periods had a different influence on the accuracy of the SLA extraction. The albedo thresholds (>0.6) for distinguishing bare ice and snow were higher in 2019 and 2020, indicating that there was more snow cover on the glacier surface in the corresponding period. Although the surface albedo showed a prominent trend with rising altitude, there was a certain influence on the derived SLA. Moreover, the processing steps of Landsat images will cause some uncertainty in the glacier SLA, such as Landsat ETM+ strips repaired and the snow-ice mixed pixels near the snowline. For (2) and (3), the uncertainty is hard to quantify; an accurate number is not given. The correlation between the albedo derived from Landsat images and the measured albedo from the AWS indicated that the workflow for extracting the SLA from Landsat images based on the variation of the surface albedo with the altitude was reliable. It has been successfully applied to glaciers in the western Himalayan Mountains and the seven glaciers in the eastern Tianshan Mountains with a quantifiable uncertainty of less than ± 25 m [30,31]. In this study, the resulting total uncertainty of each SLA was controlled in the range of ± 30 m by error propagation.

### 5.3. Potential Impact of Albedo Variation on Glacier Melt

The glacier surface albedo is closely related to its mass balance because the albedo controls the energy budget process between the glacier surface and the atmosphere [46,47]. In this study, we investigated the relationships between the summer (from June to August)

average surface albedo values retrieved from the MOD10A1 and the annual glacier mass balance measured during 2017–2020. The mass balance used to establish the relationship here was obtained by partition statistics at an altitude of 3100–3400 m a.s.l. according to an altitude interval of 100 m. The average albedo was 0.23 in summer, and the annual mass balance was −1399 mm w.e. during 2017–2020. A significant correlation (R = −0.84) was observed at the 99% confidence level between the average annual glacier mass balance and the average surface albedo in summer, indicating that the glacier mass loss or accumulated corresponds to the decrease or increase in surface albedo (Figure 8). The linear relationship between the albedo and mass balance indicated that the 0.005 decrease in albedo led to glacier mass loss enhanced by 100 mm w.e. on the Muz Taw Glacier. It has been confirmed in previous studies [48,49] that the decrease in the albedo accelerated the glacier melt. It had been found that the summer average albedo also showed a significant relationship with the annual average mass balance for glaciers in different mountains, based on the existing observation data and previous studies. For instance, the 0.006 decrease in the albedo of the Urumqi Glacier No.1 in the Tianshan Mountains possibly caused glacier mass loss by 100 mm w.e. during 2001–2018 [23,50]. For the Laohugou glacier No.12 in the Qilian Mountains, the decrease in the surface albedo by 0.03 corresponding with the glacier melt was enhanced by 100 mm w.e. during 2001–2018 [51]. For the Xiaodongkemadi Glacier in the Tanggula Mountains, Central Tibetan Plateau, the albedo decreasing by 0.02 possibly caused glacier mass loss of 100 mm w.e. during 2001–2010 [14,52]. Although the contribution of the glacier surface albedo to the mass balance existed in different mountains distinctly, the variation of the surface albedo mentioned above cannot be neglected in the estimation of glacier melt.

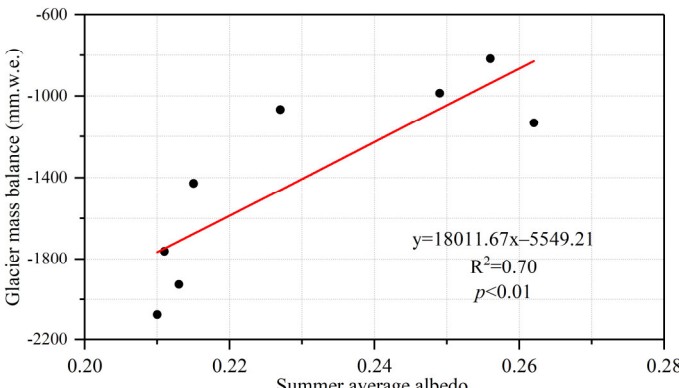

**Figure 8.** Relationships between summer average albedo and annual glacier mass balance for the Muz Taw Glacier.

### 5.4. The Impact of Air Temperature and Precipitation Variation on Snowline Altitude Changes

The intense ablation and accumulation of the Muz Taw Glacier usually occur simultaneously from June to September. Figure 9 shows the changes in air temperature and precipitation from June to September over the 2011–2021 period. During the study period, the air temperature increased slightly at a rate of 0.008 °C·a$^{-1}$, with an average air temperature of 8 °C. The total precipitation showed a downward trend, and no significant trend was observed in solid precipitation. The SLA was extremely sensitive to changes in air temperature and precipitation variables. In this study, the sensitivity of the SLA to climate variables was evaluated by establishing a linear regression between SLA anomalies, air temperature anomalies (Figure 10a), and solid precipitation anomalies (Figure 10a). As shown in Figure 10a, an increase or decrease in air temperature results in an increase or decrease in glacier SLA. There was a positive correlation between the air temperature and the SLA, and the correlation coefficient was 0.78 with an error less than 1%. Notably, although the correlation between the solid precipitation and the SLA was not statistically significant at the 95% confidence level, the SLA clearly decreased or increased with the

increased or decreased solid precipitation, as shown in Figure 10b. Regarding total precipitation, no significant correlation was found. On the whole, the results demonstrated that air temperature exerted the primary influence on the SLA changes in the Muz Taw Glacier. The linear analysis between the air temperature and the SLA indicated that the glacier SLA increased by approximately 62 m when the average temperature increased by 1 °C from June to September.

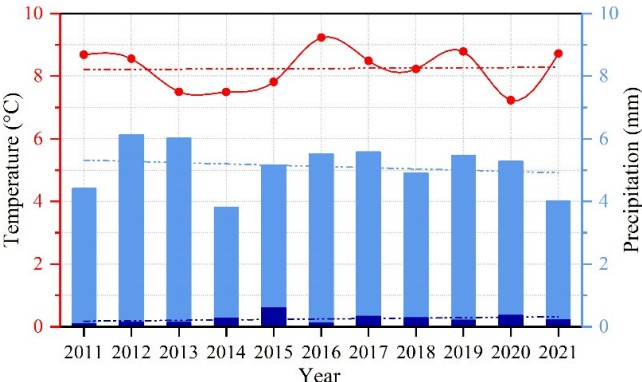

**Figure 9.** Variations in the average air temperature and the precipitation from June to September. The red dots represent the air temperature, the light blue bars represent the total amount of precipitation, the dark blue bars represent the amount of solid precipitation, and the dotted line represents the linear trend.

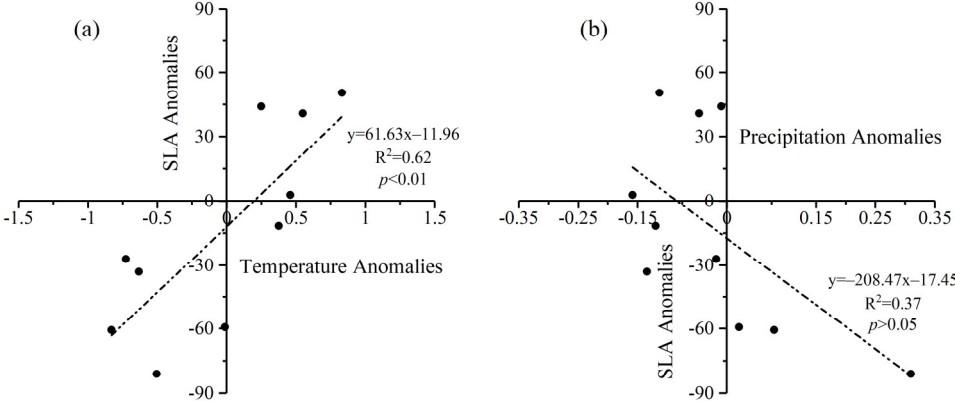

**Figure 10.** Average snowline altitude anomalies plotted versus air temperature anomalies (**a**) and solid precipitation anomalies (**b**) on the Muz Taw Glacier during 2011–2021.

### 5.5. Comparison of SLA with Other Glaciers in High Mountain Asia

There were significant spatio-temporal differences of the glacier SLA when comparing the Muz Taw Glacier with other typical glaciers in High Mountain Asia (HMA), according to the previous studies [30,31,53–56] (Figure 11). For instance, the average SLA of ~3440 m a.s.l. and ~3445 m a.s.l. was derived for the Maliy Aktru Glacier and Leviy Aktru Glacier in the Altai Mountains, with an upward trend during 2000–2016. The SLA of the TS.Tuyuksuyskiy Glacier and the Urumqi Glacier No.1 in the Tianshan Mountains increased approximately at the rate of 7.4 m·a$^{-1}$ and 9.47 m·a$^{-1}$, respectively, during 2000–2016, with an average SLA of ~4063 m and ~4471 m a.s.l., respectively. For the Qiyi Glacier in the Qilian Mountains, the average SLA during 1989–2018 increased by ~340 m at a rate of 11.9 m·a$^{-1}$. During 2000–2016, the SLA of the Xiaodongkemadi Glacier in the central Tibetan Plateau and the Parlung No.94 Glacier in the southeast Tibetan Plateau increased at a rate of ~6.4 m·a$^{-1}$ and ~7.3 m·a$^{-1}$, respectively, with the average SLA of ~5640 m and ~5470 m a.s.l., respectively. The average SLA of the Chhota Shigri Glacier and the Mera Glacier located in the western and central Himalayas Mountains showed a significant

upward trend, with the average SLA of ~5340 m and ~6200 m, respectively, during the same period.

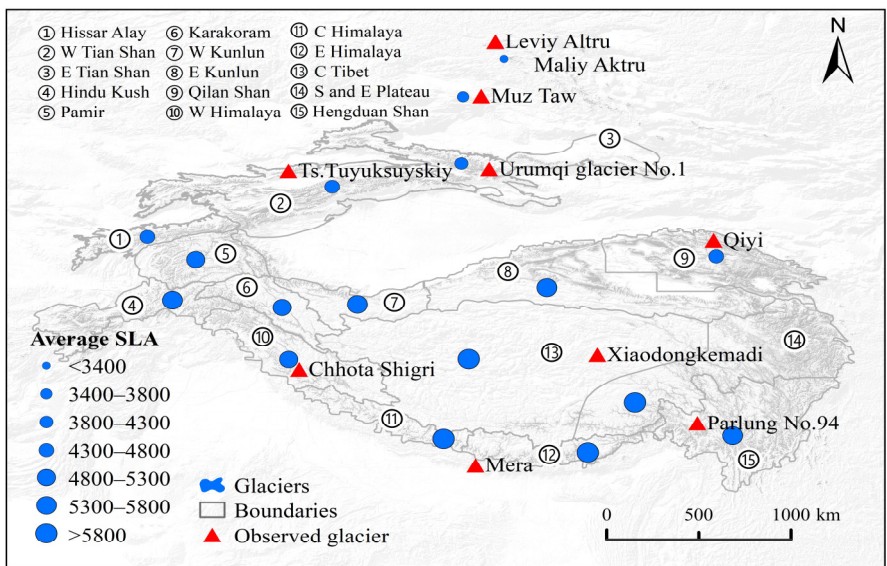

**Figure 11.** Spatial distribution of the average SLA in High Mountain Asia over the period of 2000–2021. The blue dots represent the average SLA of the glacier in the different regions.

On the whole, the SLA of typical glaciers studied in HMA all showed an upward trend over the study period. However, the average SLA in different mountain regions showed significant variations. The glacier SLA in the Altai Mountains increased at a rate of ~2.9 m·a$^{-1}$ during 2000–2019, with an average SLA of ~3126 m a.s.l. During 2001–2016, linear regression analysis revealed that the average SLA of glaciers in the Tianshan Mountains usually increased along with the time. The SLA of glaciers in the eastern and western Tianshan Mountains showed an average increasing rate of ~5.2 m·a$^{-1}$ and ~4.7 m·a$^{-1}$, respectively, with an average SLA of ~4143 m a.s.l. and ~4365 m a.s.l., respectively. The average SLA of glaciers in the central and southeastern Tibetan Plateau increased by about ~36 m and ~92 m per decade, respectively, during the same period, with an average SLA of ~5945 m a.s.l. and ~5817 m a.s.l., respectively. The average SLA of glaciers in the Qilian Mountains was ~4779 m a.s.l. during 1988–2018, with a downward trend from western to eastern. The linear analysis indicated that the overall mean glacier SLA in the Qilian Mountains increased at a rate of ~7.8 m·a$^{-1}$. During 2000–2019, the average glacier SLA in the Karakoram Mountains was ~5253 m a.s.l., with an increased rate of ~7.4 m·a$^{-1}$. For the glaciers in the Himalayas, the average glacier SLA in the western part was ~5260 m a.s.l. However, the glacier SLA revealed a declining trend of approximately ~7.4 m·a$^{-1}$ during 2000–2019. The average glacier SLA was relatively higher than ~5800 m a.s.l. in the central and eastern Himalayas, with an increasing trend of ~6.3 m·a$^{-1}$. Except for the glaciers in the western Himalayas Mountains, the SLA showed an upward trend with remarkable spatial distribution characteristics. The glacier SLA increased with rising latitude, meaning that the glacier SLA on the south of HMA was generally higher than that on the north. The glacier SLA on the eastern side of the mountain was usually higher than that on the west, with a similar trend in the changing rate. This spatial distribution phenomenon was probably related to regional climatic conditions, such as temperature, precipitation, airstream, etc. Generally, the higher temperature and the lower solid precipitation led to a faster upward movement of the glacier SLA. The climate changes controlled by atmospheric circulations were the primary drivers for determining the glacier SLA changes [52,55,56].

## 6. Conclusions

The glacier surface albedo of the Muz Taw Glacier was investigated based on Landsat images, MOD10A1 products, and in situ measurements of the surface albedo from the AWS. The surface albedo along the glacier main flowline was also extracted from the Landsat images at the end of the melt season. The threshold between ice and snow was classified according to the change in the surface albedo along the glacier main flowline at 50 m intervals with rising altitude. The SLA was eventually determined by 5 m-interval contours. The results showed that the higher correlation of >0.7 with a risk of error lower than 5% was observed between the surface albedo derived from remote sensing images and in situ-measured albedo from the AWS. A slight upward trend with an average annual increase of 0.24% for the annual average albedo from MOD10A1 products was shown from 2011 to 2021. A unimodal seasonal variation in the albedo was demonstrated, with a downward trend from January to August and an upward trend from August to December. The minimum albedo customarily appeared between late June and early September. The spatial distribution of the glacier surface albedo derived from Landsat images was not entirely dependent on changes in altitude due to the dramatic effects of the topography and the glacier surface conditions. A remarkable correlation between the glacier surface albedo and the glacier mass balance existed, indicating that the decrease of the surface albedo can accelerate the glacier melt.

The SLA extracted based on albedo changes with rising altitude showed an upward trend during 2011–2021, ranging from 3375 m a.s.l. in 2015 to 3535 m a.s.l. in 2021, with a variation of 160 m and an increased rate of ~6.9 m·a$^{-1}$. The correlation coefficient of 0.78 with an error of less than 1% between the air temperature and SLA indicated that the temporal variations in the average SLA of the glacier were mainly attributed to air temperature changes. On the whole, the SLA of most glaciers in HMA showed an upward trend over the period of 2000–2021 on the temporal scale, and the SLA of glaciers in HMA increased with the rising latitude on the spatial scale.

The glacier surface albedo and SLA were considered to be the key parameters and important indicators for simulating and evaluating glacier accumulation or melt. The surface albedo and SLA on the Muz Taw Glacier were estimated in this study. However, the relationship between the surface albedo of an individual glacier and negative mass balance or retreat was not accurately quantified by coupled observations or modeling. There is no quantitative description of the relationship between SLA and ELA due to the limitations of observation data. In future works, the constraints of surface albedo on mass balance, snowmelt runoff, and hydrological models should be further focused on, and the relationship between SLA and ELA should be explored more deeply to provide a reference for related studies.

**Author Contributions:** Data curation, H.L.; Writing—original draft, F.Y.; Supervision, P.W. All authors have read and agreed to the published version of the manuscript.

**Funding:** This research was jointly funded by the Youth Innovation Promotion Association of the Chinese Academy of Sciences (Y2021110), the National Natural Science Foundation of China (41771077), and the State Key Laboratory of Cryospheric Science (SKLCS-ZZ-2022).

**Data Availability Statement:** The data that support the findings of this study are available from the corresponding author upon reasonable request.

**Conflicts of Interest:** The authors declare no conflict of interest.

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
