# Peer review of "Surface Albedo and Snowline Altitude Estimation Using Optical Satellite Imagery and In Situ Measurements in Muz Taw Glacier, Sawir Mountains"

_remotesensing, doi:10.3390/rs14246405_

Round 1
Reviewer 1 Report
In this study, MuzTaw Glacier in Sawir Mountains were used to investigate the variations of glacier surface albedo and the contribution of surface albedo to glacier melt. The study is of scientific value and contributes to the understanding of the impact of climate change on the cryosphere. I think the article can be published after correcting the following issues.
1- Table I, giving the percentage of clouds for each of the selected LANDSAT images.
2-Line 134-136. “The moderate resolution imaging Spectroradiometer product (MOD10A1) with the high temporal resolution of 1 day was used in this study to verify the accuracy of glacier surface albedo derived from Landsat satellite…”The spatial resolution of MODIS is smaller than LANDSAT, and coarse resolution data is not suitable for validating high resolution imagery.
3- The study period is short, why choose only 2011-2020?
4- In section 3.2.2. Is the method of snow line extraction original to the authors or is it using the results of previous research. If it is a previous study borrowed, appropriate references need to be given.
5- The figure caption of Figure 2a is incomplete, please give the specific meaning of each element. Also the trend and significance of the albedo please label the graph.
6- In Figure4, Why does the albedo show an increase and then decrease with increasing altitude?
Author Response
Please see the attachment. “remotesensing-2032354_response to reviewers' comments“

Reviewer 2 Report
This paper has a great deal of merit. The authors have done a careful job with their analysis. However there are a number of areas wherein the paper falls short. Many grammar and sentence structure issues are evident, and other areas need scientific clarification. I have suggested changes in grammar, etc., but there are many additional problems that I did not identify because there are so many. Please have an English expert read the final paper.
This is minor but please be consistent throughout the paper. Either write Muz Taw Glacier or Muz Taw glacier. Muz Taw Glacier, with an upper case G, is preferred.
My additional comments are provided below.
Line
15 plus mention that MOD10A1 comes from the Moderate-resolution Imaging Spectroradiometer (MODIS) sensor.
36 I don’t think that the mass loss affects climate change. It is affected by climate change.
37 delete the word ‘meanwhile’
45-46 change to “… glaciers in the Sawir Mountains have experienced accelerated retreat…”
50 depend should read ‘depends’
62 spell out the MODIS acronym here
62 delete the words ‘in the meantime’
62-63 this sentence doesn’t make very much sense
66 Delete “Therefore” and replace with “A”
67 delete ‘optical images’
71 delete the word ‘rising’
74 Do you mean Landsat satellite images?
81 delete ‘deeply’
83 delete ‘variation’
86 East should read ‘east’
89 glacierswith should read ‘glaciers with’
91-92 change to: “glaciers exist in the Sawir…”
93 change exists to ‘exist’
101 change the word during to ‘from’
114 spell out the acronym TM
122 delete the word ‘the’ in front of snowfall
126 clouds should say ‘cloud’
127 delete occurring resulted and replace it with ‘resulting’
162 the heading implies that you have meteorological data but you do not; you have modeled data, so you might want to consider changing the title of the heading to ‘Meteorological Data from ERA-5”
171 change were to ‘was’
175-176 ‘due to instrument fault’ does not make sense; please fix
268 change when the to ‘as’
277 clouds should say ‘cloud’
281 needs a space ‘from 2018’
282 delete the word ‘the’
292-295 please check, but I think this has already been stated at least once earlier
303 change to: “…with an increase in altitude, which…”
310 change to: “…which will be discussed in detail in section 4.3.”
322 change turning to the word ‘inflection’
329-333 change all of the words ‘was’ to the word ‘is”
349 Discussions should read ‘Discussion’
351 change will be to ‘is”
353 change the word in to the word ‘for’
361 I suggest to start a new paragraph here
369-372 please re-phrase for clarity
390 change to “…and has good prospects for…”
411 the words Western and East should start with lower case
420 Delete “It should be noted that”
415 the first sentence of this section is good; this is a key point of the paper
423 please be more specific about what is being correlated to achieve 0.84
445 a 10-year period is not considered climate; you really need 30 years before you can start to consider it as climate or climate change – refer to Fig 9
460-462 this is a key point but it is hidden
460 change radical to ‘primary’
Figure 9 – the dark blue doesn’t show up well; please use a different color
484 should MERA be all in caps?
488 delete as time goes by and replace it with ‘over the study period.’
525-528 This value of agreement seems so good that it is almost unbelievable. Can you back that up?
546 sentence that starts at the end of this line and goes onto the next page needs to be re-phrased for clarity
Author Response
Reviewer 2:
Comments and Suggestions for Authors
This paper has a great deal of merit. The authors have done a careful job with their analysis. However, there are a number of areas where in the paper falls short. Many grammar and sentence structure issues are evident, and other areas need scientific clarification. I have suggested changes in grammar, etc., but there are many additional problems that I did not identify because there are so many. Please have an English expert read the final paper.
Reply: The authors thank the reviewer for their comments on our manuscript here. We have made major improvements in the revised manuscript. Especially, we have invited a native English expert to polish the English language of the revised manuscript.
This is minor but please be consistent throughout the paper. Either write Muz Taw Glacier or Muz Taw glacier. Muz Taw Glacier, with an upper case G, is preferred.
Reply: Yes, change to “Muz Taw Glacier” throughout revised manuscript.
My additional comments are provided below.
Line
15 plus mention that MOD10A1 comes from the Moderate-resolution Imaging Spectroradiometer (MODIS) sensor.
Reply: Yes, we reorganized this sentence marked in yellow in the revised manuscript Line 14-16. The text had been modified in detail as follows:
“In this study, the surface albedo of Muz Taw Glacier was derived from Landsat images with a spatial resolution of 30 m and the Moderate Resolution Imaging Spectroradiometer albedo products (MOD10A1) with a temporal resolution of 1 day from 2011 to 2021,”
36 I don’t think that the mass loss affects climate change. It is affected by climate change.
Reply: Yes. We rephrased this sentence in the revised manuscript Line 36-38. The text had been modified in detail as follows:
“The accelerated glaciers mass loss significantly affects river runoff, regional water cycle, as well as ecology and social economy, and increases the possibility of regional water resources disasters [4-6].”
37 delete the word ‘meanwhile’
Reply: Yes. Deleted.
45-46 change to “… glaciers in the Sawir Mountains have experienced accelerated retreat…”
Reply: Yes. Corrected.
50 depend should read ‘depends’
Reply: Yes. Corrected.
62 spell out the MODIS acronym here
Reply: Yes. Corrected.
62 delete the words ‘in the meantime’
Reply: Yes. Deleted.
62-63 this sentence doesn’t make very much sense
Reply: Yes. Deleted.
66 Delete “Therefore” and replace with “A”
Reply: Yes. Replaced.
67 delete ‘optical images’
Reply: Yes. Deleted.
71 delete the word ‘rising’
Reply: Yes. Deleted.
74 Do you mean Landsat satellite images?
Reply: Yes. Corrected.
81 delete ‘deeply’
Reply: Yes. Deleted.
83 delete ‘variation’
Reply: Yes. Deleted.
86 East should read ‘east’
Reply: Yes. Corrected.
89 glacierswith should read ‘glaciers with’
Reply: Yes. Corrected.
91-92 change to: “glaciers exist in the Sawir…”
Reply: Yes. Corrected.
93 change exists to ‘exist’
Reply: Yes. Corrected.
101 change the word during to ‘from’
Reply: Yes. Corrected.
114 spell out the acronym TM
Reply: Yes, Corrected. We corrected in the revised manuscript Line 114: “...Landsat-7 Enhanced Thematic Mapper Plus (ETM +)...”
122 delete the word ‘the’ in front of snowfall
Reply: Yes. Deleted.
126 clouds should say ‘cloud’
Reply: Yes. Corrected.
127 delete occurring resulted and replace it with ‘resulting’
Reply: Yes. Replaced.
162 the heading implies that you have meteorological data but you do not; you have modeled data, so you might want to consider changing the title of the heading to ‘Meteorological Data from ERA-5”
Reply: Yes. Corrected. We corrected in the revised manuscript Line 160. The title has been changed to “Meteorological Data from ERA-5”
171 change were to ‘was’
Reply: Yes. Corrected.
175-176 ‘due to instrument fault’ does not make sense; please fix
Reply: Yes. We rephrased the sentence marked in yellow in the revised manuscript Line 173-175. The text has been modified in detail as follows:
“It should be noted that the Landsat ETM+ instrument with a scan line corrector malfunctioned in 2003, resulting in data gaps. These data gaps were repaired by neighboring pixels of the image gaps”.
268 change when the to ‘as’
Reply: Yes. Corrected.
277 clouds should say ‘cloud’
Reply: Yes. Corrected.
281 needs a space ‘from 2018’
Reply: Yes. Corrected.
282 delete the word ‘the’
Reply: Yes. Deleted.
292-295 please check, but I think this has already been stated at least once earlier
Reply: Yes. We deleted this sentence and added some text in the revised manuscript Line 293-300: The text has been modified in detail as follows:
“Overall, the glacier surface albedo would continue to decline throughout the melt sea-son and eventually reach its minimum. Although the albedo continued to decline be-tween May and June, the value remained above 0.6. And then the glacier surface albedo showed an accelerated decline in July, and stabilized at the minimum ordinarily between July and August, the value remained below 0.5. The surface albedo increased again with the snowfall and fresh snow accumulation since late August or the beginning of September. These phenomena had also been founded on Urumqi Glacier No.1 in TianShan [23].”
303 change to: “…with an increase in altitude, which…”
Reply: Yes. Corrected.
310 change to: “…which will be discussed in detail in section 4.3.”
Reply: Yes. Corrected.
322 change turning to the word ‘inflection’
Reply: Yes. Corrected.
329-333 change all of the words ‘was’ to the word ‘is”
Reply: Yes. Corrected.
349 Discussions should read ‘Discussion’
Reply: Yes. Corrected.
351 change will be to ‘is”
Reply: Yes. Corrected.
353 change the word in to the word ‘for’
Reply: Yes. Corrected.
361 I suggest to start a new paragraph here
Reply: Yes. Corrected.
369-372 please rephrase for clarity
Reply: Yes, we rephrased the sentence marked in yellow in the revised manuscript Line 373-378. The text has been modified in detail as follows:
“The greater deviation in surface albedo derived from MOD10A1 mainly stemmed from two aspects: (1) Spectral properties. The snow albedo was higher in the visible spectrum and lower in the shortwave and infrared band; (2) Mixed pixels. Average from the 500 m pixel resolution of MOD10A1 corresponds to the point-scale AWS data, and the biases induced by mixed pixels at the edge of and outside was not considered.”
390 change to “…and has good prospects for…”
Reply: Yes. Corrected.
411 the words Western and East should start with lower case
Reply: Yes. Corrected.
420 Delete “It should be noted that”
Reply: Yes. Deleted.
415 the first sentence of this section is good; this is a key point of the paper
Reply: Yes. Thanks!
423 please be more specific about what is being correlated to achieve 0.84
Reply: Yes. We rephrased the part in the revised manuscript Line 432-435. The text has been modified in detail as follows:
“A significant correlation (R=-0.84) was observed at the 99% confidence level between the average annual glacier mass balance and the average surface albedo in summer, indicating that the glacier mass loss or accumulated corresponds to the decrease or increase in surface albedo (Figure 8).”
445 a 10-year period is not considered climate; you really need 30 years before you can start to consider it as climate or climate change – refer to Fig 9
Reply: Yes. We rephrased the title in the revised manuscript Line 454. The text had been modified in detail as follows:
“The Impact of Temperature and Precipitation Variation on Snowline Altitude Changes”.
460-462 this is a key point but it is hidden
Reply: Yes. The sensitivity of SLA to air temperature was estimated by the linear relationship between SLA anomalies and air temperature anomalies. We rephrased the part and added some text to explain this calculation process in the revised manuscript Line 455-474. In addition, we marked the linear formula between variables in Figure 10 as follows.
“The intense ablation and accumulation of the Muz Taw Glacier usually occurs simultaneously from June to September. Figure 9 shows the changes in air temperature and precipitation from June to September over the 2011-2021 period. During the study period, the air temperature increased slightly at a rate of 0.008°C·a−1, with an average air temperature of 8°C. The total precipitation showed a downward trend, and no sig-nificant trend was observed in solid precipitation. The SLA was extremely sensitive to changes in air temperature and precipitation variables. In this study, the sensitivity of SLA to climate variables was evaluated by establishing a linear regression between SLA anomalies and air temperature anomalies (Figure 10a) and solid precipitation anomalies (Figure 10a), respectively. As shown in Figure 10a, an increase or decrease in air temperature results in an increase or decrease in glacier SLA. There was a positive correlation between the air temperature and SLA, and the correlation coefficient was 0.78 with an error less than 1%. Notably, although the correlation between the solid precipitation and SLA was not statistically significant at the 95% confidence level, the SLA decreased or increased with the increased or decreased solid precipitation clearly, as shown in Figure 10b. Regarding total precipitation, no significant correlation was found. On the whole, the results demonstrated that air temperature exerted the pri-mary influence on the SLA changes in Muz Taw Glacier. The linear analysis between the air temperature and SLA indicated that the glacier SLA increased by approximately 62 m when the average temperature increased by 1°C from June to September.”
Figure 10. Average snowline altitude anomalies plotted versus air temperature anomalies (a) and solid precipitation anomalies (b) on Muz Taw Glacier during 2011-2021, respectively.
460 change radical to ‘primary’
Reply: Yes. Corrected.
Figure 9 – the dark blue doesn’t show up well; please use a different color
Reply: Yes. Corrected as follows.
Figure 9. Variations in the average air temperature, and the precipitation from June to September. The red dots represent the air temperature, the light blue bars represent the total amount of precipitation, the dark blue bars represent the amount of solid precipitation, and the dotted line represents the linear trend.
484 should MERA be all in caps?
Reply: Yes. Changed to “Mera”.
488 delete as time goes by and replace it with ‘over the study period.’
Reply: Yes. Corrected.
525-528 This value of agreement seems so good that it is almost unbelievable. Can you back that up?
Reply: Yes. In “5.1 Uncertainty estimation of Glacier Surface Albedo”, we describe in detail the value of agreement between the surface albedo derived from remote sensing images and in situ measured albedo from AWS, including the correlation, absolute error, root mean square error and mean absolute error. And we give the specific value of agreement in order to make the meaning of this sentence clearer, in the revised manuscript Line 537-541. The text in detail as follows:
Line 537-541: The results showed that the higher correlation of >0.7 with a risk of error lower than 5% was observed between the surface albedo derived from remote sensing images and in situ measured albedo from AWS. A slight upward trend with an average annual increase of 0.24% for the annual average albedo from MOD10A1 products was shown from 2011 to 2021.
5.1 Uncertainty estimation of Glacier Surface Albedo: Landsat vs. in situ measured albedo: The correlation was 0.95 at the 95% confidence level between the Landsat-derived and the measured albedo. The Absolute Error (AE) ranged from 0.01 to 0.07, with the Root Mean Square Error (RMSE) of 0.038 and Mean Absolute Error (MAE) of 0.03 (Figure 7a).
MOD10A1 vs. in situ measured albedo: The correlation between MOD10A1 albedo and measured albedo was 0.75 at 99% centroid level and the difference ranged from -0.18 to 0.25, with the RMSE of 0.13 and MAE of 0.05, respectively (Figure 7b).
546 sentence that starts at the end of this line and goes onto the next page needs to be rephrased for clarity
Reply: Yes, we rephrased the sentence marked in yellow in the revised manuscript Line 559-561. “However, the relationship between surface albedo of an individual glacier and negative mass balance or retreat was not accurately quantified by coupled observations or modeling.”
The attach has been uploaded.
Reviewer 3 Report
Review for: “Surface Albedo and Snowline Altitude Estimation Using Optical Satellite Imagery and in-situ measurements in Muz Taw Glacier, Sawir Mountains”
Glacier mass balance is controlled by glacier surface albedo. At the end of the melt season, the glacier snowline altitude might indicate changes in glacial mass balance. This study calculated and evaluated glacier mass balance, glacier surface albedo and SLA.
This is an interesting work. The manuscript is generally very good and the topic seems to present very interesting results for readers. I suggest a "Minor" revision before possible consideration of the application in remote sensing. My comments are listed as:
1. The abstract was beefily describe the paper with quantitative results and overall background. It is required to mention a conclusion sentence in the end of it.
2. L64 to 66: Please revise the meaning of the sentence
3. Please check Figure 1 as figure 1b is not a part of figure 1a (Lat is different in both figures)
4. Figure 2 is too small.
Author Response
Reviewer 3:
Comments and Suggestions for Authors
Review for: “Surface Albedo and Snowline Altitude Estimation Using Optical Satellite Imagery and in-situ measurements in Muz Taw Glacier, Sawir Mountains”
Glacier mass balance is controlled by glacier surface albedo. At the end of the melt season, the glacier snowline altitude might indicate changes in glacial mass balance. This study calculated and evaluated glacier mass balance, glacier surface albedo and SLA.
This is an interesting work. The manuscript is generally very good and the topic seems to present very interesting results for readers. I suggest a "Minor" revision before possible consideration of the application in remote sensing. My comments are listed as:
Reply: The authors thank the reviewer for their comments on our manuscript here. We will further address these comments and improvements in the replies to the specific comments.
- The abstract was beefily describe the paper with quantitative results and overall background. It is required to mention a conclusion sentence in the end of it.
Reply: Yes, we added some text in the Line 29-30. The text has been added in detail as follows:
“This study improved our understanding into ablation process and mechanism of the Muz Taw Glacier.”
- L64 to 66: Please revise the meaning of the sentence
Reply: Yes, we rephrased the sentence marked in yellow in the revised manuscript Line 64-67. The text has been modified in detail as follows:
“Although Landsat images are more susceptible to cloud cover and shadow, temporal resolution of which is lower than that of MODIS images, the combination of Landsat images and MODIS can be used to obtain the surface albedo with a higher spatial resolution and a longer time series.”
- Please check Figure 1 as figure 1b is not a part of figure 1a (Lat is different in both figures)
Reply: Yes. Figure 1b is a part of figure 1a. The problem is due to the different projections, which has been revised. The modified Figure1 is as follows.
- Figure 2 is too small.
According to the comments from the reviewer 2 and 3, Figure 2 had been modified as follows,

Round 2
Reviewer 2 Report
I have no further comments.